# Offline Multi-Agent Reinforcement Learning with Sequential Score Decomposition

## Abstract

Offline multi-agent reinforcement learning (MARL) faces significant challenges due to distribution shift issues, exacerbated by the high dimensionality of joint actions and complex joint behavior policy distributions. While existing methods often focus on independent learning or offline value decomposition with conservative value estimation, they may still lead to out-of-distribution (OOD) joint actions and reduced performance. This is primarily due to the lack of exploration opportunity and implicit policy dependencies in offline settings. To address these challenges, we propose an offline policy decomposition method incorporating joint policy regularization constraints. Our approach utilizes a diffusion generative model to capture the joint behavior policy, followed by a decomposition of the extracted score function. This decomposition is then used to regularize individual policies in a decentralized manner. Experimental results demonstrate that our method achieves SOTA on continuous control tasks in standard offline MARL benchmarks.

## 1 Introduction

In recent years, Multi-Agent Reinforcement Learning (MARL) has demonstrated remarkable progress in addressing complex decision-making problems that necessitate high-quality coordination among multiple entities. Significant achievements have been realized in challenging domains such as DOTA, soccer simulations, StarCraft II, and AI-driven economic models (Zhang et al., 2021a; Guo et al., 2023; Chen et al., 2021; Mannion et al., 2016; Zheng et al., 2020; Berner et al., 2019; Ma et al., 2024). However, the online learning paradigm often impedes the application of MARL algorithms to broader real-world scenarios, particularly when simulation environments fail to accurately capture real-world complexities or when real-world exploration entails inherent risks and substantial costs. For example, creating a simulation environment that comprehensively replicates market economics or allowing MARL algorithms to explore government incentive policies and gather market feedback for policy updates presents formidable challenges (Zheng et al., 2022; Wang et al., 2024; Gao et al., 2024). Consequently, offline MARL has emerged as a promising paradigm (Formanek et al., 2023; 2024). By leveraging existing datasets to develop effective strategies without necessitating direct environmental interaction during training, offline MARL potentially bridges the gap between simulated and real-world applications.

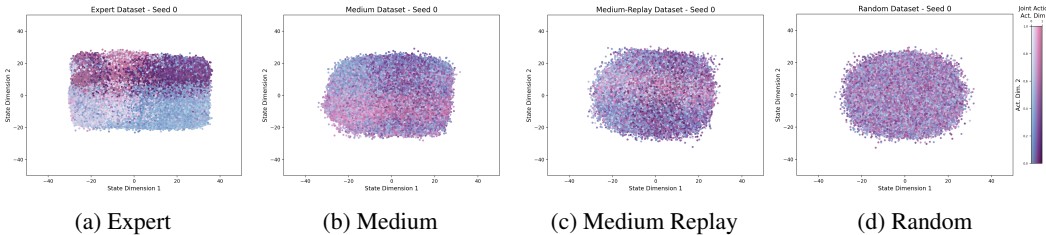

| (a) Expert | (b) Medium | (c) Medium Replay | (d) Random |

Figure 1: Visualization of `MAMujoco` (Peng et al., 2021) datasets across different quality datasets. With the decreasing of dataset quality, the distribution shows more multi-modal and symmetry.

A primary challenge in offline MARL is addressing the distribution shift problem that arises from the discrepancy between the learned policy and the data collection policy. Current approaches to offline

MARL primarily fall into two categories: independent learning methods (Yang et al., 2021; Jiang & Lu, 2021; Ma & Wu, 2023; Shao et al., 2023) and offline value decomposition paradigms (Wang et al., 2023a;c). These approaches typically adopt the conservatism principle from single-agent offline reinforcement learning (Levine et al., 2020; Prudencio et al., 2023), employing pessimistic estimation of each agent's value function. However, these methods have inherent limitations: independent learning lacks effective coordination mechanisms, while offline value decomposition methods are susceptible to selecting out-of-distribution joint actions during decentralized execution.

Our visualization analysis of standard offline MARL datasets `MAMujoco` (Peng et al., 2021) reveals a critical factor contributing to the performance gap between existing methods: the presence of multiple optimal policy combinations, analogous to the classic multiple Nash Equilibria (NE) or equilibrium selection problem in game theory (Tian et al., 2023; Franzmeyer et al., 2024). As illustrated in Fig. 1, this phenomenon results in extremely complex joint policy distributions, posing unique challenges for offline MARL algorithms. For instance, in SMAC cooperative tasks, teams can achieve victory through various, equally efficient strategies, leading to a multi-modal joint policy space. This complexity manifests even in simple XOR coordination games, where the presence of multiple global optima can render many existing algorithms ineffective. Individual-Global Maximization (IGM) based methods may erroneously select OOD joint actions, while independent training approaches often struggle to maintain coordination (Matsunaga et al., 2023). Alarmingly, even behavioral cloning on expert trajectory datasets can lead to suboptimal policy combinations, failing to capture the intricate interdependencies among agents.

To elucidate the complexity of these challenges, consider a simple coordination game where two agents must synchronize to achieve an optimal joint action. Even in such a rudimentary structure, the presence of multiple global optima can render many existing algorithms ineffective. For instance, IGM-based methods may erroneously select out-of-distribution (OOD) joint actions, while decentralized training approaches, operating under the assumption of fixed policies for other agents, may encounter similar issues. More disconcertingly, even behavioral cloning on expert trajectory datasets can lead to the selection of OOD joint actions. Moreover, it is fundamentally impossible to recover a joint policy distribution with multiple equilibria through any pairwise individual policies.

In this work, we propose a novel offline MARL algorithm, named **Offline MARL with Sequential Score Decomposition** (OMSD), that leverages advanced Diffusion models to get accurate score functions of the joint behavior policies, and decompose the joint score functions into coordinated individual score functions for each agent's policy regularization.

Our method encompasses the following key contributions: (1) Introduction of Diffusion Models to accurately capture complex joint policy distributions, offering superior expressiveness compared to traditional Variational Auto-Encoder (VAE) or mixture of Gaussian distributions in representing the intricate (Wang et al., 2023b), multimodal distributions inherent in MARL environments. (2) Development of a novel score function decomposition method, enabling the extraction of individual policies from joint policies while maintaining overall coordination. This approach overcomes the limitations of conventional policy decomposition methods and effectively handles symmetric and near-symmetric policy distributions. (3) Theoretically, we demonstrate the efficacy of our method in handling complex MARL environments, particularly highlighting its advantages in addressing symmetric and near-symmetric policy distributions. Our analysis indicates that the approach effectively mitigates OOD behaviors while preserving the underlying task's shared reward structure. Empirical results corroborate these findings, with our method significantly outperforming existing value-based and policy-based approaches across multiple tasks in standard offline MARL benchmarks provided in OMAR (Pan et al., 2022).

## 2 PRELIMINARIES

### 2.1 PARTIALLY OBSERVABLE STOCHASTIC GAME

A partially observable stochastic game (POSG; Hansen et al., 2004) is defined as a tuple:

$$\langle \mathcal{X}, \mathcal{S}, \{\mathcal{A}^i\}_{i=1}^n, \{\mathcal{O}^i\}_{i=1}^n, \mathcal{P}, \mathcal{E}, \{\mathcal{C}^i\}_{i=1}^n \rangle,$$

where $n$ is the number of agents, $\mathcal{X}$ is the agent space, $\mathcal{S}$ is a finite set of states, $\mathcal{A}^i$ is the action set for agent $i$, $\mathbfcal{A} = \mathcal{A}^1 \times \mathcal{A}^2 \times \cdots \times \mathcal{A}^n$ is the set of joint actions, $\mathcal{P}(s'|s, \mathbf{a})$ is the state transition

probability function, $\mathcal{O}^i$ is the observation set for agent $i$, $\mathcal{O} = \mathcal{O}^1 \times \mathcal{O}^2 \times \cdots \times \mathcal{O}^n$ is the set of joint observations, $\mathcal{E}(\boldsymbol{o}|s)$ is the emission function, and $\mathcal{R}^i : \mathcal{S} \times \mathcal{A} \times \mathcal{S} \to \mathbb{R}$ is the reward function for agent $i$. The game progresses over a sequence of stages called the *horizon*, which can be finite or infinite. This paper focuses on the episodic infinite horizon problem, where each agent aims to minimize the expected discounted cumulative cost.

In a cooperative POSG (Song et al., 2020b), the relationship between agents $x$ and $x'$ is given by:

$$\forall x \in \mathcal{X}, \forall x' \in \mathcal{X} \setminus \{x\}, \forall \pi_x \in \Pi_x, \forall \pi_{x'} \in \Pi_{x'}, \frac{\partial \mathcal{R}^{x'}}{\partial \mathcal{R}^x} \geqslant 0,$$

where $\pi_x$ and $\pi_{x'}$ are policies in the policy spaces $\Pi_x$ and $\Pi_{x'}$, respectively. This means there is no conflict of interest among any pair of agents. The paper addresses the fully cooperative POSG, also known as the decentralized partially observable Markov decision process (Dec-POMDP; Bernstein et al., 2002), where all agents share the same global cost at each stage, i.e., $\mathcal{R}^1 = \mathcal{R}^2 = \cdots = \mathcal{R}^n$. The optimization goal for Dec-POMDP is defined as:

$$\min_{\Psi} \sum_{i=1}^{n} \sum_{t=0}^{\infty} \mathbf{E}_{s_0 \sim p_0, \boldsymbol{o} \sim \mathcal{E}, a \sim \boldsymbol{\pi}_\Psi} \left[ \gamma^t r_{t+1}^i \right], \tag{1}$$

where $\Psi := \{\psi^i\}_{i=1}^n$ are the parameters of the approximated policies $\pi_{\psi^i}^i : \mathcal{O}^i \to \mathcal{A}^i$, and $\boldsymbol{\pi}_\Psi := \prod_{i=1}^n \pi_{\psi^i}^i$ is the joint policy of all agents. Here, $\gamma$ is the discount factor, $p_0$ is the initial state distribution, and $r_{t+1}^i$ is the reward received by agent $i$ at timestep $t+1$ after taking action $a_t^i$ in observation $o_t^i$. In the offline setting, we only have a static dataset of transitions $\mathcal{D} = (o_t^m, a_t^m, o_{t+1}^m, r_t^m)_{m=1}^{nk}$, where $k$ is the number of transitions for each agent.

## 2.2 DIFFUSION PROBABILISTIC MODELS

Diffusion probabilistic models (Sohl-Dickstein et al., 2015; Ho et al., 2020) are a likelihood-based generative framework designed to learn data distributions $q(\boldsymbol{x})$ from offline datasets $\mathcal{D} := \boldsymbol{x}^i$, where $i$ indexes individual samples (Song, 2021). A key feature of these models is the representation of the (Stein) score function (Liu et al., 2016), which does not require a tractable partition function.

The model's discrete-time generation procedure involves a forward noising process, defined as $q(\boldsymbol{x}_{k+1}|\boldsymbol{x}_k) := \mathcal{N}(\boldsymbol{x}_{k+1}; \sqrt{\tilde{\alpha}_k}\boldsymbol{x}_k, (1-\tilde{\alpha}_k)\boldsymbol{I})$, at diffusion timestep $k$. This is paired with a learnable reverse denoising process, $p_\theta(\boldsymbol{x}_{k-1}|\boldsymbol{x}_k) := \mathcal{N}(\boldsymbol{x}_{k-1}|\mu_\theta(\boldsymbol{x}_k, k), \Sigma_k)$, where $\mathcal{N}(\mu, \Sigma)$ represents a Gaussian distribution with mean $\mu$ and variance $\Sigma$. The variance schedule is defined by $\alpha_k \in \mathbb{R}$. In this framework, $\boldsymbol{x}_0 := \boldsymbol{x}$ corresponds to a sample in $\mathcal{D}$, and $\boldsymbol{x}_1, \boldsymbol{x}_2, \ldots, \boldsymbol{x}_{K-1}$ are latent variables, with $\boldsymbol{x}_K \sim \mathcal{N}(\boldsymbol{0}, \boldsymbol{I})$ for appropriately chosen $\tilde{\alpha}_k$ values and a sufficiently large $K$.

Starting with Gaussian noise, samples are iteratively generated through a series of denoising steps. The training of the denoising operator is guided by an optimizable and tractable variational lower bound, with a simplified surrogate loss proposed in (Ho et al., 2020):

$$\mathcal{L}_{\text{denoise}}(\theta) := \mathbb{E}_{k \sim [1,K], \boldsymbol{x}_0 \sim q, \epsilon \sim \mathcal{N}(\boldsymbol{0},\boldsymbol{I})} \left[ \| \epsilon - \epsilon_\theta(\boldsymbol{x}_k, k) \|^2 \right]. \tag{2}$$

Here, the predicted noise $\epsilon_\theta(\boldsymbol{x}_k, k)$, parameterized by a deep neural network, approximates the noise $\epsilon \sim \mathcal{N}(\boldsymbol{0}, \boldsymbol{I})$ added to the dataset sample $\boldsymbol{x}_0$ to produce the noisy $\boldsymbol{x}_k$ in the noising process.

## 2.3 POLICY BASED OFFLINE RL

Policy based methods are successful and widely used in the offline RL algorithm community. Previous works (Nair et al., 2020) has provided the problem formulation as:

$$\max_{\pi} \mathbb{E}_{s \sim \mathcal{D}_\mu} \left[ \mathbb{E}_{a \sim \pi(s)} [Q_\phi(s, a)] - \frac{1}{\beta} \mathcal{D}_{\text{KL}} \left( \pi(\cdot|s) \| \mu(\cdot|s) \right) \right], \tag{3}$$

where $Q_\phi(s, a)$ is a neural network trained to estimate the state-action value functions $Q^\pi(s, a) := \mathbb{E}_{s_1=s, a_1 \sim \pi}[\sum_{n=1}^\infty \gamma^n r(s_n, a_n)]$ under the current policy $\pi$, and $\beta$ is temperature coefficient to control how far the learned policy derive from the behavior policy $\mu$. The closed form solutions for this optimization problem (3) has been proved as

$$\pi^*(a \mid s) = \frac{1}{Z(s)} \mu(a \mid s) \exp\left( \beta Q_\phi(s, a) \right), \tag{4}$$

where $Z(s)$ is the partition function. The following problem is to efficiently distill the optimal policy into a parameterized policy $\pi_\theta$. The common practice are minimizing the KL-divergence between $\pi_\theta$ and $\pi^*$ with either forward-KL or reverse-KL. Although the optimal policy may be multi-modal, meaning it has multiple equivalent policy mode distributions, it is not necessary to express every policy mode explicitly during execution. Therefore, it is a suitable choice to leverage the natural of mode-seeking characteristic in reverse-KL and capture only one mode in the parameterized policy with a simple distribution like Gaussian policy or deterministic policy.

**Lemma 1** (Behavior-Regularized Policy Optimization (BRPO), Wu et al. (2019)). *In policy-based offline RL, given an optimal policy $\pi^*$ and a parameterized policy $\pi_\theta$, the policy regularization learning objective with reverse KL-divergence can be written as,*

$$\min_\theta \mathbb{E}_{s\sim\mathcal{D}_\mu} \underbrace{D_{KL}\left[\pi_\theta(\cdot|s)\|\pi^*(\cdot|s)\right]}_{\text{Reverse KL}} \Leftrightarrow \max_\theta \underbrace{\mathbb{E}_{s\sim\mathcal{D}_\mu, a\sim\pi_\theta} Q_\phi(s,a) - \frac{1}{\beta} D_{KL}\left(\pi_\theta(\cdot|s)\|\mu(\cdot|s)\right)}_{\text{Behavior-Regularized Policy Optimization}}. \quad (5)$$

# 3 METHODOLOGY

In this section, we introduce our algorithm, namely OMSD, that addresses key challenges in offline MARL through sequential score decomposition techniques derived from pre-trained diffusion models. In subsection 3.1, we begin by analyzing the limitations of existing policy-based offline MARL methods based on BRPO, focusing on independent learning and Centralized Training with Decentralized Execution (CTDE) frameworks. To deal with these limitations, we propose an unbiased score decomposition method for coordinated policy updates in subsection 3.2.

## 3.1 CHALLENGES FOR POLICY DECOMPOSITION IN OFFLINE MARL

First, we analyze the failure modes of policy-based methods in offline MARL, providing a detailed and comprehensive understanding of the current research gap. Following common modeling approaches in online MARL settings, we formulate the optimization objectives in offline MARL using two different policy learning methods: independent learning and the CTDE framework.

**Policy-based Offline MARL with Independent Learning.** We begin our analysis with BRPO-Ind, a fundamental case under the independent learning paradigm. Generally, independent learning methods decompose MARL problems into multiple autonomous single-agent RL processes. This is a robust approach widely adopted in both online and offline MARL algorithms that has demonstrated stable performance across many tasks. Specifically, each agent independently updates its critic and actor components using shared team reward signals, while modeling its individual behavior policy $\mu_i(a_i|s)$. For BRPO-Ind, we propose the following proposition:

**Proposition 1.** *Consider a fully-cooperative game with n agents. Under the independent learning framework, the optimal individual policy of each agent is:*

$$\pi_i^*(a_i \mid s) = \frac{1}{Z(s)} \mu_i(a_i \mid s) \exp\left(\beta_i Q^i(s, a_i)\right),$$

*where $\mu_i$ and $Q^i$ are individual behavior policy and Q-value function of agent $i$, respectively. With Lemma 1, the learning objective of BRPO-Ind is:*

$$\mathcal{L}_{Ind} = \min \sum_{i=1}^{n} \mathbb{E}_{s\sim\mathcal{D}_\mu} D_{KL}\left[\pi_{\theta_i}\|\pi_i^*\right] \Leftrightarrow \max \sum_{i=1}^{n} \mathbb{E}_{s\sim\mathcal{D}_\mu, a_i\sim\pi_{\theta_i}} Q^i(s, a_i) - \frac{1}{\beta} D_{KL}\left[\pi_{\theta_i}\|\mu_i\right]. \quad (6)$$

By taking the gradient of Equation (6) with respect to each agent's policy parameters, we obtain:

$$\nabla_{\theta_i}\mathcal{L}_{Ind} = \mathbb{E}_{s\sim\mathcal{D}^\mu}\left[\nabla_{a_i}Q^i(s,a_i)\big|_{a_i=\pi_{\theta_i}} + \frac{1}{\beta}\underbrace{\nabla_{a_i}\log\mu_i(a_i\mid s)\big|_{a_i=\pi_{\theta_i}(s)}}_{=-\epsilon_i^*(a_t|s,t)/\sigma_t|_{t\to 0}}\right]\nabla_{\theta_i}\pi_{\theta_i}(a_i|s), \quad (7)$$

where $\epsilon_i^*\left(a_t \mid s, t\right)$ represents the score function of behavior policy $\nabla_{a_i}\mu_i(a_i|s)$ (Song et al., 2020a).

**Policy-based Offline MARL with CTDE Learning.** In the CTDE framework, the centralized training process typically leverages other agents' actions and global states information to learn optimal joint strategies. The joint policy is then appropriately decomposed to obtain executable individual policies. For value-based methods, the IGM (Individual-Global-Max) principle is often relied upon to decompose the centralized critic $Q_{tot}(s, a_1, a_2, \ldots, a_n)$ into local value estimation networks $\hat{Q}_i(s, a_i)$ suitable for individual policy execution. In policy-based MARL methods, such as FOP (Zhang et al., 2021b) and AlberDICE (Matsunaga et al., 2023), the IGO (Individual-Global-Optimal) decomposable assumption is typically used to directly extract individual policies.

However, considering that limited coverage of offline data can lead to biased estimates of joint value functions and the OOD joint action selection problem, decomposing such biased value functions without the ability to interact with the environment to obtain new data would further increase the bias in individual value functions. Therefore, we suggest avoiding the decomposition of the global value function and instead directly decomposing the joint optimal policy to obtain individual execution policies. Based on this approach, we designed the BRPO-CTDE algorithm as follows.

**Proposition 2.** *Consider a fully cooperative game with n agents. In centralized learning process, the optimal joint policy is derived as*

$$\pi^*(\boldsymbol{a} \mid s) = \frac{1}{Z(s)} \mu(\boldsymbol{a} \mid s) \exp\left(\beta Q^{tot}(s, \boldsymbol{a})\right), \tag{8}$$

*where $\boldsymbol{a}$ represents the joint actions and $Q^{tot}$ represents the global state-action value function. With Lemma 1 and the IGO principle, the learning objective for each agent becomes*

$$\mathcal{L}_{CTDE}^i = \min_{\theta_i} \mathbb{E}_{s \sim \mathcal{D}_\mu} D_{\mathrm{KL}}[\pi_\theta(\cdot \mid s) \| \pi^*(\cdot \mid s)] \tag{9}$$

$$= \min_{\theta_i} \mathbb{E}_{s \sim \mathcal{D}^\mu, \boldsymbol{a} \sim \pi_\theta(\cdot \mid s)} Q^{tot}(s, \boldsymbol{a}) - \frac{1}{\beta} D_{\mathrm{KL}}[\pi_\theta(\boldsymbol{a} \mid s) \| \mu(\boldsymbol{a} \mid s)]$$

Then we can get the gradient of Equation (9) with respect to each agent's policy parameters as:

$$\nabla_{\theta_i} \mathcal{L}_{CTDE}^i = \mathbb{E}_{s \sim \mathcal{D}^\mu, a^{-i} \sim \pi_{-i}} \left[ \nabla_{a_i} Q^{tot}(s, \boldsymbol{a}) \big|_{a_i = \pi_{\theta_i}(\cdot \mid \boldsymbol{s}), a_{-i} = \pi_{\theta_{-i}}(\cdot \mid \boldsymbol{s})} \right.$$
$$\left. + \frac{1}{\beta} \nabla_{a_i} \log \mu(\boldsymbol{a} \mid s) \big|_{a_i = \pi_{\theta_i}(s)} \right] \nabla_{\theta_i} \pi_{\theta_i}(a_i \mid s). \tag{10}$$

Equations (7) and (10) reveal that the gradients in offline policy-based MARL consist of Q-value gradients and behavior policy regularization terms. Unfortunately, this structure introduces significant challenges for joint policy update. An obvious problem arises in the coordination of Q-value gradients part. In offline MARL, the absence of online data collection severely limits the ability to adjust policies through newcoming experiences. Consequently, the direction of Q-value gradients heavily depend on the coverage of the offline datasets, potentially missing optimal gradient directions. This issue is further exacerbated by the non-convex nature of the value function. Even when individual agents' policies satisfy local improvement, the gradients of the joint policy may still lead to suboptimal directions due to misalignment of individual gradients (Kuba et al., 2022; Pan et al., 2022).

Admittedly, the CTDE frameworks can slightly alleviate the Q-value gradients problem by directly providing local gradients of the joint Q-function to each agent. However, the multi-modal property of the joint behavior policy greatly chanllenge the regularization process. In online learning, due to the coordinated updates between policies, the joint policy typically exhibits a unimodal nature, which can be decomposed as the product of all agents' individual policies. This property no longer holds in offline datasets that contain multiple joint policies of equal quality. Applying this assumption in such cases often leads to distribution shift and suboptimal policy regularization. We demonstate this phenomenon in the following proposition. The proof is provided in Appendix B.3.

**Proposition 3** (**Distribution Shift of Joint Behavior Policy**). *Consider a fully-cooperative n-players game with a single state and action space $\mathcal{A} = [0,1]^n$. Let $\pi^*$ be the optimal joint policy with two optimal modes: $\mathbf{a}_1 = (1, ..., 1)$ and $\mathbf{a}_2 = (0, ..., 0)$. Let $\hat{\pi}$ be a factorized approximation of $\pi^*$ such that $\hat{\pi}(\mathbf{a}) = \prod_{i=1}^n \hat{\pi}_i(a_i)$, where each $\hat{\pi}_i$ is learned independently. Then we have each $\hat{\pi}_i$ converges to Uniform$(\{0, 1\})$. The reconstruction of joint policy $\hat{\pi}$ exhibits $2^n$ modes, each with probability $2^{-n}$. The total variation distance between $\pi^*$ and $\hat{\pi}$ is:*

$$\delta_{TV}(\pi^*, \hat{\pi}) = 1 - 2^{1-n} \tag{11}$$

*As $n \to \infty$, $\delta_{TV}(\pi^*, \hat{\pi}) \to 1$, indicating a severe distribution shift.*

Besides, when we leverage a high-capacity generative model, such as diffusion models, to represent the behavior policy distributions, we cannot distill the regularization term as shown in Equation (10) that $\nabla_{a_i} \log \mu(a|s) = \nabla_{a_i} \pi * \nabla_\pi \log \mu(a|s)$. Here, $\nabla_\pi \log \mu(a|s)$ represents the score function of the joint behavior policy, while $\nabla_{a_i} \pi$ is the partial gradient of the joint policy with respect to agent $i$. The primary difficulty lies in accurately calculating $\nabla_{a_i} \pi$, as the offline joint policy may not be easily factorizable into individual agent policies.

This proposition underscores a critical distinction between online and offline MARL, emphasizing the multi-modal nature of offline joint policies and the inappropriateness of naive factorization in these settings. While previous work like AlberDICE discussed similar issues from a value-based OOD joint action selection perspective, our work identifies this problem from a policy-based OOD joint action distribution shift viewpoint, offering a novel perspective on the challenges in offline MARL.

### 3.2 SEQUENTIAL SCORE DECOMPOSITION

To address these challenges, we need to focus on modeling joint behavior policy with powerful generative models and developing effective decomposition methods. Under the BRPO framework, when we have trained a perfect generative model for behavior policy, we can naturally distill the score functions as policy regularization. The decomposition of joint policies thus translates to the decomposition of score functions.

The key challenge is to keep joint actions within the support sets of joint behavior policies while obtaining individual regularization terms for each agent. Naive factorization in the KL divergence of the joint policy only constrains update directions towards individual behavior policy distributions, failing to guarantee joint update directions stay close to the joint behavior policy distribution. With independent learning or IGO-based CTDE frameworks, score terms become either biased and uncoordinated or intractable.

To address this, inspired by coordination descent and Multi-agent Transformer (MAT, Wen et al. (2022)), we adopt sequential policy decomposition as $\boldsymbol{\mu}(\boldsymbol{a}|s) = \Pi_i^n \mu(a_i|s, a^{i-})$, where $a^{i-}$ represents the joint actions of prefix agents of agent $i$. The KL divergence of the joint policy becomes $D_{KL}(\Pi_i^n \pi_i(a_i|s)||\Pi_i^n \mu(a_i|s, a^{i-}))$ and the corresponding regularization for each agent is $\hat{\epsilon}_i = -\sigma_t \nabla_{a_i} \log \mu(a_i|s, a^{i-})$.

By plugging this score term into the BRPO framework, we can propose our new algorithm as OMSD (Offline MARL with sequential score decomposition). The global information of the joint behavior policy distribution is transferred as local information of relative action distributions, providing fine-grained regularization and stable numerical computation. The objective loss becomes

$$\mathcal{L}_{OMSD}^i = \min_{\theta_i} \mathbb{E}_{s \sim \mathcal{D}_\mu} D_{\mathrm{KL}}[\pi_\theta(\cdot \mid s)||\pi^*(\cdot \mid s)] \tag{12}$$

$$= \min_{\theta_i} \mathbb{E}_{s \sim \mathcal{D}^\mu, \boldsymbol{a} \sim \pi_\theta(\cdot|s)} Q^{tot}(s, \boldsymbol{a}) - \frac{1}{\beta} D_{\mathrm{KL}} \left[ \pi_{\theta_i}(\cdot \mid \boldsymbol{s}) \pi_{\theta_{-i}}(\cdot \mid \boldsymbol{s}) || \mu_i(\cdot \mid \boldsymbol{s}, a_{i-}) \boldsymbol{\mu}_{-i} \right],$$

where $\boldsymbol{\mu}_{-i}$ represents all other sequential decomposed behavior policies. The gradient of loss objective w.r.t each agent's policy parameter is:

$$\nabla_{\theta_i} \mathcal{L}_{OMSD}^i = \mathbb{E}_{s \sim \mathcal{D}^\mu, a^{-i} \sim \pi_{-i}} \left[ \nabla_{a_i} Q^{tot}(s, \boldsymbol{a}) \big|_{a_i = \pi_{\theta_i}(\cdot|\boldsymbol{s}), a_{-i} = \pi_{\theta_{-i}}(\cdot|\boldsymbol{s})} \right.$$

$$\left. + \frac{1}{\beta} \nabla_{a_i} \log \mu_i(\cdot \mid \boldsymbol{s}, a_{i-}) \big|_{a_i = \pi_{\theta_i}(s)} \right] \nabla_{\theta_i} \pi_{\theta_i}(a_i|s). \tag{13}$$

It is important to note that in OMSD, the sequential conditional distribution is only used in pretrained diffusion models to obtain score functions for policy updates. Unlike online algorithms that design sequential execution policies to address non-stationarity, our approach distills information about other agents' actions into the score function solely for individual policy training. This design ensures that during execution, each agent's policy remains independently executable based only on local observations, in contrast to methods like MAT that require sequential action selection. Thus, OMSD-SSD maintains the benefits of coordinated learning while preserving simultaneous decision-making capabilities in deployment. This approach provides greater flexibility in policy execution, allowing for adaptability in various multi-agent scenarios.

Table 1: Evaluation rewards after convergence for the toy example

| BRPO-Ind | BRPO-JAL | BRPO-FAC | BRPO-SSD |
|----------|----------|----------|----------|
| $0\pm1$ | $1\pm0$ | $0\pm1$ | $1\pm0$ |

## 4 PRACTICAL ALGORITHM

The OMSD methods contain a two-stages training process: 1) pretraining diffusion models and joint action critic on the dataset and make score decomposition, and 2) injecting decomposed scores as the policy regularization terms into the critic and derive deterministic policies for execution. The resulting OMSD algorithm is presented in Algorithm 1.

The basic workflow of OMSD follows the idea of SRPO (Chen et al., 2024) by extending the single agent learning process into multi-agent process, where the unbiased score decomposition methods proposed in section 3.2 are plugged-in to avoid the uncoordination policy updated. Specifically, as we take the joint critic and individual score regularization, all the agents share the copies of a pretrained common joint action Q-networks $Q_{tot}$ and keep individual pre-trained behavior diffusion models to extract the score regularization. This is a common setup in multi-agent reinforcement learning, such as MADDPG. Besides, each agent maintains a deterministic policy as the actor network, which bypasses the heavy iterative denoising process of diffusion models to generate actions and enjoy the fast decision-making speed.

---

**Algorithm 1** OMSD Algorithm

1: Initialize parameters.
2: // Critic training (IQL)
3: **for all** critic training steps **do**
4:     Pretrain a centralized joint Critic $Q^{tot}$
5: **end for**
6: // Behavior training
7: **for all** gradient step **do**
8:     Pretrain sequential diffusion models proposed in Sec. 3.2.
9: **end for**
10: // Policy extraction
11: **for all** gradient step **do**
12:     Update $\theta \leftarrow \theta + \alpha\nabla_\theta L_{OMSD}(\theta)$ (13)
13: **end for**

---

## 5 EXPERIMENTS AND RESULTS

In this section, we evaluate our method on the designed bandit example in section 3.1 and challenging high-dimensional continuous control tasks `MAMujoco`. Specifically, in `MAMujoco`, each part of a robot is modeled as an independent agent and learn optimal motions by cooperating with each other.

**Datasets.** In the bandit example, we design the 2-dimensional joint action distribution as a 2-Gaussian Mixed Model with mean values $\mu_0 = [0.8, 0.8]$, $\mu_1 = [-0.8, -0.8]$ and variance $\sigma_0 = \sigma_1 = 0.3$. As a multi-modal distribution, sequential factorization is necessary for avoiding distribution shifting problems in this case. We collected 1,000,000 joint action samples to construct the bandit dataset. In `MAMujoco`, we use 2-agent HalfCheetah as the experimental environments, where the dataset comes from the widely used datasets in offline MARL papers provided by OMAR (Pan et al., 2022).

**Baselines.** In the bandit example, we focus on the performance of three classical MARL frameworks, i.e., BRPO-Ind, BRPO-FAC, and BRPO-JAL. By comparing OMSD with these clean baselines, we can check the learning process and analyze the performance significantly. In `MAMujoco` experiments, our chosen benchmarks include two mainstream of state-of-the-art baselines in offline MARL: independent learning algorithms (OMAR (Pan et al., 2022)), and CTDE learning algorithms (MA-CQL (Jiang & Lu, 2021) and MAIGM (Wang et al., 2023a)). We also consider diffusion-based offline MARL techniques, such as DOM2 (Li et al., 2023). These diffusion methods provide more details about the influence of score decomposition.

### 5.1 BANDIT EXAMPLE PERFORMANCE

We first evaluate OMSD on the 2d-bandit as shown in Fig. 2 to illustrate the drawbacks of existing offline MARL methods. The maximize of rewards in this environment is 1, which can be achieve by selecting the joint actions either {1, 1} or {-1, -1}. We show the results in Table 5.1.

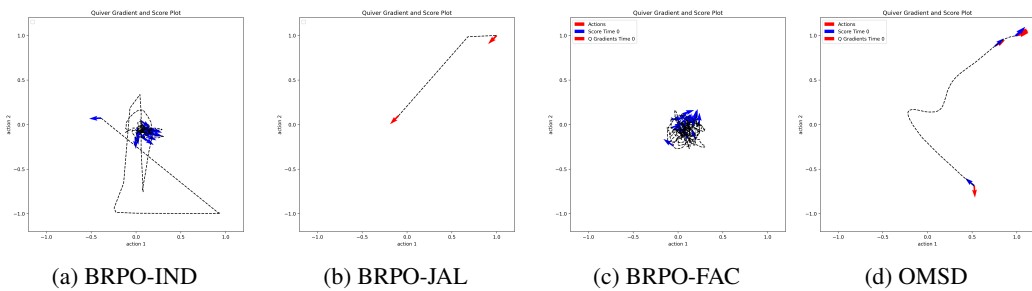

| (a) BRPO-IND | (b) BRPO-JAL | (c) BRPO-FAC | (d) OMSD |

Figure 2: Illustration of update trajectories in bandit example.

Table 2: Evaluation unnormalized scores of `MAMujoco` benchmarks. We report mean $\pm$ standard deviation of algorithm performance across 5 random seeds at the last $10\%$ training steps.

| Dataset | OMAR | MA-CQL | MAIGM | DOM2 | BRPO-Ind | BRPO-FAC | OMSD (Ours) |
|---|---|---|---|---|---|---|---|
| Expert | 2963.8±410.5 | 2722.8±1022.6 | 3383.61±552.67 | 3676.6±248.1 | 3636.2±22.5 | 3788.4±24.1 | **3881±48** |
| Medium | 2797.0±445.7 | 963.4±316.6 | **3608.13±237.37** | 2851.2±145.5 | 2744.6±19.6 | 2376.4±20.4 | 2853.3±90.9 |
| Medium-Replay | 1674.8±201.5 | 1216.6±514.6 | 2504.70±83.47 | 2564.3±216.9 | 2462.4±68.4 | 894.5±33.8 | **2757.7±185.6** |
| Random | -0.9±0.1 | -0.1±0.2 | **2948.46± 518.89** | 799.8±143.9 | -243.3±22.8 | -46.9±48.2 | 64.7±32.4 |

Our experimental results demonstrate a comparative study on the performance of various algorithms, of which the OMSD cornered a unique position boasting similar efficacy to joint action learning algorithms. This is remarkable, as both the independent learning methods and the naive factorization CTDE based methods stumbled in selecting the Out-of-Distribution (OOD) joint actions wherein $\{1, -1\}$ and $\{-1, 1\}$ are commonly noted.

Besides, compared to the previous research in discrete Matrix Game (Matsunaga et al., 2023), this problem becomes more severe in continuous tasks. It is expected that using less expressive behavior models will further reduce performance, as the behavioral policy cannot accurately capture the complex multi-modal data distribution (Wang et al., 2023b). These results strongly suggest that our decomposition is effective to guarantee the joint policy constraint and coordination in offline MARL.

To further corroborate our analysis and illustrate the challenges faced by current offline MARL approaches, we present a 2-dimensional visualization of gradient directions during the learning process . This toy example compares Joint-Action Learning (JAL), which treats MARL as a single RL problem with a large joint action space and serves as a theoretical upper bound, with Independent Learning (IND) and Centralized Training Decentralized Execution (CTDE) approaches. The results align with our theoretical analysis: IND exhibits miscoordination among independent Q-values, potentially leading to out-of-distribution (OOD) actions, as evidenced by the divergent gradient directions in 2a. CTDE, while addressing some non-stationarity issues, struggles to accurately represent the gradient of the joint distribution with respect to individual agent policies, resulting in suboptimal convergence as shown in 2c. In contrast, JAL achieves superior performance by considering the full joint action space 2b, highlighting the importance of properly handling joint policies in offline MARL. Our algorithm OMSD can overcome these challenges and converge to the optimal joint actions 2d.

## 5.2 MULTI-AGENT MUJOCO PERFORMANCE

We further evaluated our algorithm on more complex continuous control tasks in the `MAMujoco` suite. Table 2 demonstrates the performance of OMSD in the multi-agent HalfCheetah-v2 environments across various datasets. Our algorithm consistently outperforms both independent learning methods and diffusion-based methods like DOM2 without data augmentation across main datasets.

In the expert dataset tasks, we observed that the maximum evaluation rewards of OMSD and DOM2 during training both approach an upper bound of approximately 3900.0, with indistinguishable performance. We speculate that this represents the performance ceiling for that particular dataset. For the medium, medium-expert, and random datasets, OMSD achieves superior performance, reaching approximately 131% of OMAR's performance, and 103% of DOM2's performance on the medium

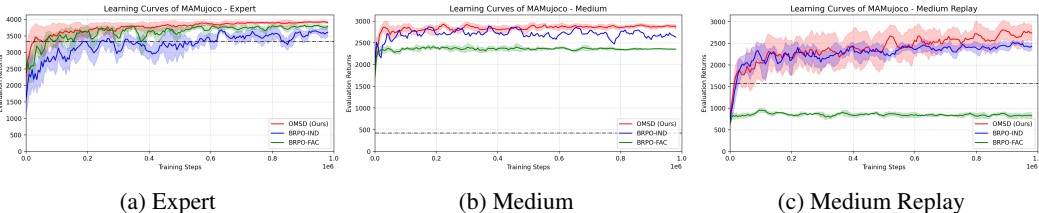

(a) Expert           (b) Medium           (c) Medium Replay

Figure 3: Learning curves of OMSD and OMSD w/o sequential decomposition in HalfCheetah-v2.

and the medium-replay datasets. The performance of OMSD is weak on random datasets, as well as DOM2 and OMAR, indicating its limitations in handling diverse and unstructured data scenarios with low data quality. Considering that in real-world scenarios we are more likely to encounter diverse datasets of moderate or near-optimal quality, rather than datasets composed purely of random data, our method still demonstrates great potential for application in these domains.

In scenarios with limited data coverage, where accurate value function estimation and decomposition is challenging, the behavioral policy proves to be a more reliable constraint. This holds true provided that an appropriate policy regularization decomposition is performed, as in our approach. Our findings suggest that OMSD's sequential score decomposition strategy effectively addresses the challenges inherent in offline MARL, particularly in environments with varying data quality and complexity.

## 5.3 ABLATION STUDY

**Does Score Decomposition Methods Matter?** To investigate the impact of our proposed SSD method, we conducted an ablation study across four `MAMujoco` datasets: HalfCheetah expert, medium, medium-replay, and random. We compared our OMSD approach against two baselines: BRPO-IND and BRPO-FAC, which is revised from BRPO-CTDE with factorization assumptions. Both baselines utilized independently pretrained diffusion models for score distillation and policy regularizations. The dot lines represent the absolute average rewards of the training datasets.

Figure 3 presents the learning curves for each method across the four datasets over 3 random seeds. Notably, OMSD consistently outperforms both BRPO-IND and BRPO-FAC across all datasets, demonstrating the effectiveness of our proposed sequential decomposition approach. The significant performance gap between BRPO-SEQ and the baselines validates our hypothesis that naive policy factorization in offline MARL leads to gradient conflicts and performance degradation.

**Hyperparameter** As policy-based offline methods are sensitive to the degree of behavior regularization, we investigate the influence of this hyperparameter on performance. We vary the regularization parameter $\beta$ across the set $\{0.005, 0.01, 0.05, 0.1, 0.2, 0.5\}$. We find that the performance of OMSD and baselines are sensitive to the regularization hyperparameter. The best performance settings of OMSD for each dataset vary from $\{0.01, 0.05, 0.1, 0.15\}$ respectively.

## 6 RELATED WORKS

**Offline MARL.** Early research in offline MARL mainly made efforts to extend the pessimistic principles from offline single-agent RL. For example, MAICQ (Yang et al., 2021) and MABCQ (Jiang & Lu, 2021) extended the pessimistic value estimation such as CQL to multi-agent and discuss the extrapolation error under exponential increasing dimension of joint actions space problem. Furthermore, OMAR (Pan et al., 2022) dealt with the local optima in independent learning paradim with zero-th order optimization. Motivated by this, CFCQL (Shao et al., 2023) further improved OMAR with counterfactual value estimation to avoid over-pessimistic value estimation. Recently, MACCA (Wang et al., 2023c) and OMIGA (Wang et al., 2023a) has incorporated causal credit assignment technique and the IGM principle into the offline value decomposition process to enhance the credit assignment. In SIT (Tian et al., 2023), authors recognized the data-imbalance problem and handle it with reliable credit assignment techinique. On the other hand, AlberDICE (Matsunaga et al., 2023) and MOMA-PPO (Barde et al., 2023) recognized and addressed OOD joint action coordination problems with alternative best response and world model based planning. Our method aligns in this

direction and try to model complex behavior policies with diffusion models. There are also some works following the trajectory generation route, such as MAT (Wen et al., 2022), MADT (Meng et al., 2021), and MADTKD (Tseng et al., 2022). These methods are beyond our discussion scoup.

**Diffusion Models in RL.** Recently, motived by the great advantage of diffusion models, RL researchers turn to seek the possibilities of introducing diffusion models into RL area. Previous works can be typically divided into three topics: serving as planner, serving as policy, and serving for data augmentation. Our method mainly fall in the second topic. Single RL suffers multimodal and MLE fails due to mode cover. Diff-QL (Wang et al., 2023b) and SfBC (Chen et al., 2022) used diffusion model to represent the behavior policy and generate a batch of candidate actions with diffusion models, then use resampling to choose the executive actions. These methods suffer the inherent drawback of slow inference process of diffusion models. For this reason, some works tried to accelerate the sampling process of diffusion actor. EDP (Kang et al., 2024) and consistency-AC (Ding & Jin, 2023) leveraged the advanced diffusion models to accerate the action sampling in RL tasks. In offline MARL, there are few works such as MADiff (Zhu et al., 2023) and DOM2 (Li et al., 2023), which take diffusion models as a centralized planner or independent actors.

# 7    CONCLUSION

This paper studies the key challenge of the unbiased decomposition of the joint action behavior distribution in the offline MARL. We start from developing CTDE algorithms based on behavior-regularized policy optimization without value decomposition and revealed an important factor which greatly limited the policy constraint methods in offline MARL: the infactorization joint policy property in the offline datasets. Based on this, we proposed two unbiased policy decomposition methods and transfer them into the gradients of agents as the score regularization distilled from the pretrained diffusion models. The experiment demonstrates the superior of our methods and the effectiveness of policy improvement with coordinate joint action selection. One future work is to develop more precise and optimal policy decomposition methods to enhance the ability of offline MARL.

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

# Supplementary Material

## Table of Contents

## A  DETAILED ANALYSIS OF OFFLINE MARL DATASET CHARACTERISTICS

In this appendix, we provide a more comprehensive analysis of the offline MARL datasets, expanding on the observations briefly mentioned in the introduction. Our analysis focuses on the `MAMujoco` datasets from OMAR (Pan et al., 2022), a widely used benchmark in offline MARL research.

Similar to single-agent offline RL, MARL datasets exhibit multi-phase distributions that vary with dataset quality. As the quality of datasets decreases, the distributions become increasingly compound and challenging to model accurately. This phenomenon is illustrated in Figure 4, which shows the joint policy distributions for four different quality levels of `MAMujoco` datasets. This complexity in distribution is a primary reason for the weak performance of policy-based MARL methods in offline settings. Accurately representing these intricate policy distributions requires advanced generative models, which are often beyond the capabilities of current offline MARL algorithms.

An important characteristic of offline MARL datasets is the variability in policy distributions, even among datasets with similar accumulated rewards. This variability stems from the randomness in source policy training and data collection processes. To illustrate this point, we conducted an experiment comparing policy distributions from different random seeds on the same task and reward level. Figure B.2 demonstrates how different seeds can lead to distinct policy distributions despite achieving similar overall performance. This variability underscores the need for offline MARL algorithms to be robust to different policy distribution patterns.

A striking feature of many offline MARL datasets is the presence of symmetry in policy distributions. This symmetry often arises from two main sources. First, multiple Nash Equilibria (NE) in multi-agent tasks lead to multiple, equally optimal solutions. For example, in a coordination game, strategies like "both agents choose left" or "both agents choose right" may be equally effective. This leads to symmetric distributions in the collected data, as illustrated in Figure B.3(a). Second, agent role symmetry occurs in environments where agents have interchangeable roles (e.g., two identical units in SMAC). In these cases, the actions of Agent 1 and Agent 2 may be equally valid when swapped. This role symmetry manifests as symmetric patterns in the joint policy distribution.

It's important to note that in real-world offline MARL datasets, distinguishing between these two types of symmetry can be challenging. As pointed out by **?**, the source of symmetry (whether from multiple NE solutions or from interchangeable agent roles) is often indiscernible in the collected data.

The complex characteristics of offline MARL datasets, including multi-phase distributions, variability across seeds, and inherent symmetries, pose significant challenges for existing algorithms. The inadequate understanding of these dataset properties leads to poor performance in value decomposition methods. Even with a well-designed value decomposition, the complexity of policy distributions can significantly hinder performance. The inherent multi-modality in these datasets, stemming

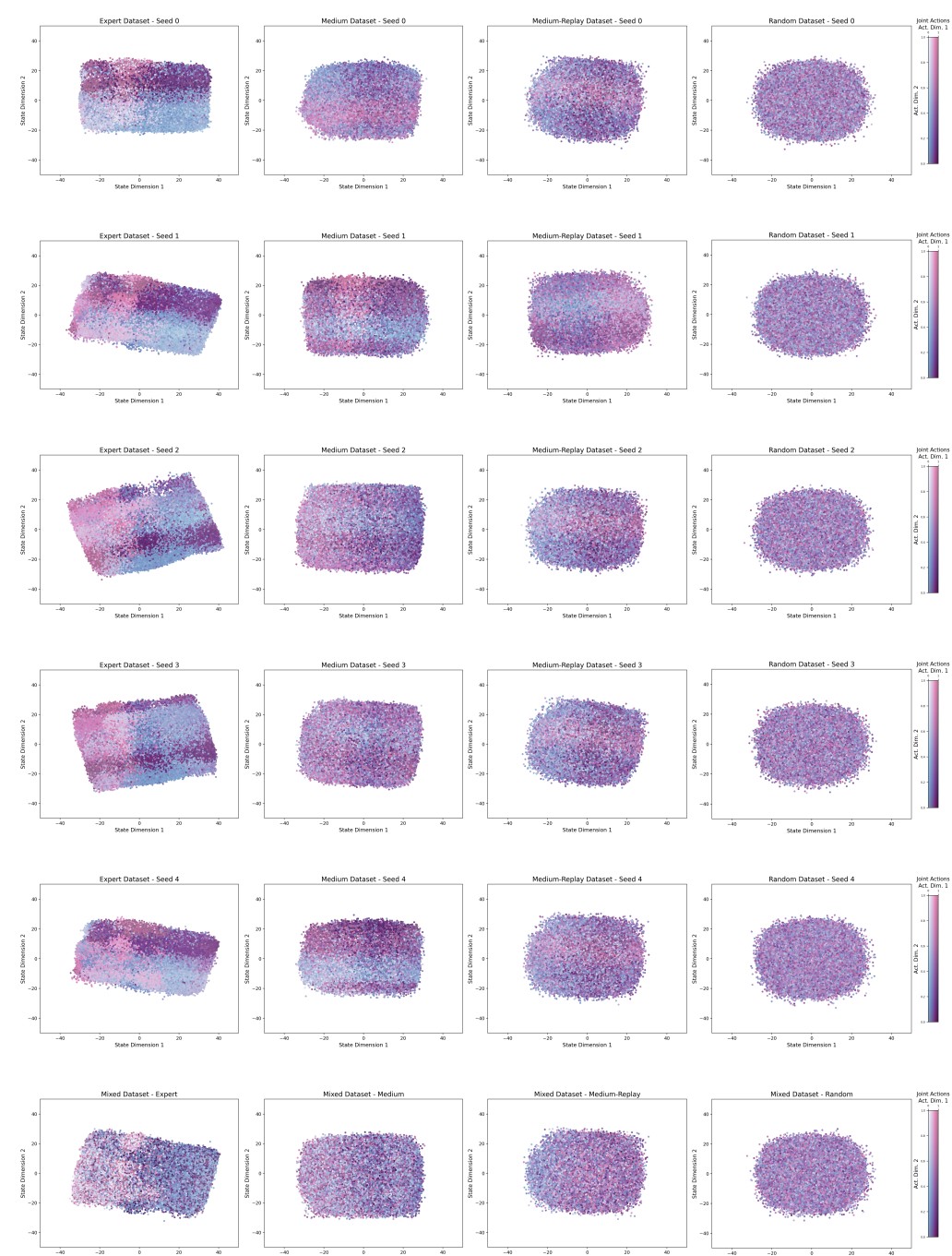

Figure 4: Visualization of `MAMujoco` datasets across all seeds and qualities.

from multiple NE and role symmetries, is a critical factor in the failure of many existing methods. Traditional approaches often struggle to capture and leverage this multi-modal nature effectively.

Moreover, the variability across datasets, even with similar reward levels, challenges the generalization capabilities of offline MARL algorithms. Methods need to be robust to different policy distribution patterns to perform well across various scenarios. This analysis underscores the need for new approaches in offline MARL that can effectively handle the unique characteristics of multi-agent datasets. Future research should focus on developing methods that can capture and exploit the com-

plex, multi-modal nature of joint policy distributions while being robust to the inherent variabilities and symmetries present in offline MARL data.

Furthermore, we constructed mixed datasets by uniformly combining complete trajectories from multiple seed datasets of the same quality level. In the last raw of Figure 4, we observed that for expert and medium datasets, while the average scores of the datasets did not change significantly, the data distribution became more complex, with more pronounced multi-modal characteristics. This better reflects real-world data collection scenarios.

# B  THEOREM DETAILS

## B.1  PROOF OF PROPOSITION 1

First, we derive the optimization objectives with independent learning framework. By decomposing the KL term in (6), we have

$$
\mathcal{L}_{Ind} = \sum_{i=1}^{n} \left( \mathbb{E}_{s\sim\mathcal{D}_\mu, a_i\sim\pi_{\theta_i}} Q^i(s, a_i) + \frac{1}{\beta}\mathbb{E}_{s\sim\mathcal{D}^\mu, a_i\sim\pi_{\theta_i}} \log\mu_i(a_i|s) + \frac{1}{\beta}\mathbb{E}_{s\sim\mathcal{D}^\mu}\mathcal{H}(\pi_i(a_i|s)) \right)
$$

where $\mathcal{H}(\pi_i(a_i|s))$ is the entropy of the agent $i$'s policy. As BRPO-Ind learns behavior policy independently, we can directly get the term $\log\mu_i(a_i|s)$ implicitly from the pretrained diffusion models of each agent.

Consider that each agent's policy is trained independently without dependency, we can derive the gradient of agent $i$ as

$$
\nabla_{\theta_i}\mathcal{L}_{Ind} = \nabla_{\theta_i}\sum_{i=1}^{n} \left( \mathbb{E}_{s\sim\mathcal{D}_\mu, a_i\sim\pi_{\theta_i}} Q^i(s, a_i) + \frac{1}{\beta}\mathbb{E}_{s\sim\mathcal{D}^\mu, a_i\sim\pi_{\theta_i}} \log\mu_i(a_i|s) + \frac{1}{\beta}\mathbb{E}_{s\sim\mathcal{D}^\mu}\mathcal{H}(\pi_i(a_i|s)) \right)
$$
(14)

$$
= \mathbb{E}_{s\sim\mathcal{D}_\mu, a_i\sim\pi_{\theta_i}} \left[ \nabla_{\theta_i} Q^i(s, a_i) + \frac{1}{\beta}\nabla_{\theta_i}\log\mu_i(a_i|s) \right]
$$
(15)

$$
= \mathbb{E}_{s\sim\mathcal{D}^\mu, a_i\sim\pi_{\theta_i}} \left[ \nabla_{\theta_i}\pi_i * \nabla_{a_i} Q^i(s, a_i) + \frac{1}{\beta}\nabla_{\theta_i}\pi_i * \nabla_{a_i}\log\mu_i(a_i|s) \right]
$$
(16)

$$
= \mathbb{E}_{s\sim\mathcal{D}^\mu, a_i\sim\pi_{\theta_i}} \left[ \nabla_{a_i} Q^i(s, a_i) + \frac{1}{\beta}\nabla_{a_i}\log\mu_i(a_i|s) \right] \nabla_{\theta_i}\pi_i.
$$
(17)

Notice that the term $\nabla_{a_i}\log\mu_i(a_i|s)$ serves as the score function of the independent behavior policy, we can further construct a surrogate loss $\mathcal{L}_{Ind}^{surr}$ and derive a practical gradient for BRPO-Ind. Our proof is mainly inspired by the following Lemma 2.

**Lemma 2** (Proposition 1 in Chen et al. (2024)). *Given that $\pi$ is sufficiently expressive, for any time $t$, any state $s$, we have*

$$
\arg\min_\pi D_{KL}[\pi_t(\cdot|s)||\mu_t(\cdot|s)] = \arg\min_\pi D_{KL}[\pi(\cdot|s)||\mu(\cdot|s)],
$$

*where both $\mu_t$ and $\pi_t$ follow the same predefined diffusion process in $q_{t_0}(x_t|x_0) = \mathcal{N}(x_t|\alpha_t x_0, \sigma_t^2 I)$, which implies $x_t = \alpha_t x_0 + \sigma_t\varepsilon$.*

The surrogate loss is

$$
L_{Ind}^{surr}(\theta_i) = \mathbb{E}_{s, a_i\sim\pi_{\theta_i}} Q(s, a_i) - \frac{1}{\beta}\mathbb{E}_{t,s}\omega(t)\frac{\sigma_t}{\alpha_t} D_{KL}[\pi_{\theta_i,t}(\cdot|s)||\mu_{i,t}(\cdot|s)].
$$
(18)

Then we can propose the practical gradient as follows.

**Proposition 4** (Practical Gradient of BRPO-Ind). *Given that $\pi_{\theta_i}$ is deterministic policy and $\epsilon_i^*$ is the optimal diffusion model of independent behavior policy $\mu_i$, the gradient of the surrogate loss (18) w.r.t agent $i$ is*

$$
\nabla_{\theta_i} L_{surr}^\pi(\theta) = \left[ \mathbb{E}_s\nabla_a Q_\phi(s, a)|_{a=\pi_\theta(s)} - \frac{1}{\beta}\mathbb{E}_{t,s}\omega(t)(\epsilon_i^*(a_{t,i}|s,t) - \epsilon_i)|_{a_{i,t}=\alpha_t\pi_{\theta_i}(s)+\sigma_t\epsilon_i} \right] \nabla_{\theta_i}\pi_{\theta_i}(s).
$$

*Proof.* The fundamental framework of the proof follows the proof process of SRPO (Chen et al., 2024), extending it to the multi-agent scenario. Based on the forward diffusion process in section 2.2, we can represent the noisy distribution of actor policy at step $t$ as

$$
\begin{aligned}
\pi_{\theta_i,t}(a_{t,i}|s) &= \int \mathcal{N}(a_{i,t}|\alpha_t a_i, \sigma_t^2 I)\pi_{\theta_i}(a_i|s)da_i \\
&= \int \mathcal{N}(a_{t,i}|\alpha_t a_i, \sigma_t^2 I)\delta(a_i - \pi_{\theta_i}(s))da_i = \mathcal{N}(a_{t,i}|\alpha_t\pi_{\theta_i}(s), \sigma_t^2 I) \quad (19)
\end{aligned}
$$

Note that $\pi_{\theta,t}(\cdot|s)$ is a Gaussian policy with expected value $\alpha_t\pi_\theta(s)$ and variance $\sigma_t^2 I$, we can simplify the surrogate training objective as

$$
\begin{aligned}
L_{Ind}^{surr}(\theta_i) &= \mathbb{E}_{s,a_i\sim\pi_{\theta_i}(\cdot|s)}Q(s,a_i) - \frac{1}{\beta}\mathbb{E}_{t,s}\omega(t)\frac{\sigma_t}{\alpha_t}D_{\mathrm{KL}}[\pi_{\theta_i,t}(\cdot|s)\|\mu_{i,t}(\cdot|s)] \\
&= \mathbb{E}_s Q(s,a_i)|_{a_i=\pi_{\theta_i}(s)} + \frac{1}{\beta}\mathbb{E}_{t,s}\omega(t)\frac{\sigma_t}{\alpha_t}\mathbb{E}_{a_{i,t}\sim\mathcal{N}(\cdot|\alpha_t\pi_{\theta_i}(s),\sigma_t^2 I)}[\log\mu_t(a_{i,t}|s) - \log\pi_{t,\theta_i}(a_{i,t}|s)]
\end{aligned}
$$

Then we can derive the gradient of this objective as follows

$$
\begin{aligned}
\nabla_{\theta_i}\mathcal{L}_{Ind}^{surr}(\theta_i) &= \nabla_{\theta_i}\mathbb{E}_{\boldsymbol{s}\sim\mathcal{D}^\mu}Q_\phi(\boldsymbol{s},a_i)|_{a_i\sim\pi_\theta^i(\boldsymbol{s})} \\
&+ \frac{1}{\beta}\mathbb{E}_{t,s}\frac{\sigma_t}{\alpha_t}\omega(t)\nabla_{\theta_i}\mathbb{E}_{\epsilon_i}\left[\log\mu_t^i(a_t^i|s) - \log\pi_t^i(a_t^i|s)\right] \\
&\quad (\text{reparameterization of } \pi_i = \alpha_t\pi_{\theta_i}(s) + \sigma_t\epsilon_i) \\
&= \nabla_{\theta_i}\mathbb{E}_{\boldsymbol{s}\sim\mathcal{D}^\mu}Q_\phi(\boldsymbol{s},a_i)|_{a_i\sim\pi_\theta^i(\boldsymbol{s})} \\
&+ \frac{1}{\beta}\mathbb{E}_{t,s,\epsilon_i}\frac{\sigma_t}{\alpha_t}\omega(t)\left[\nabla_{\theta_i}\log\mu_t^i(a_t^i|s) - \nabla_{\theta_i}\log\pi_t^i(a_t^i|s)\right] \quad (\text{chain rule}) \\
&= \nabla_{\theta_i}\mathbb{E}_{\boldsymbol{s}\sim\mathcal{D}^\mu}Q_\phi(\boldsymbol{s},a_i)|_{a_i\sim\pi_\theta^i(\boldsymbol{s})} \\
&+ \frac{1}{\beta}\mathbb{E}_{t,s,\epsilon_i}\frac{\sigma_t}{\alpha_t}\omega(t)\big[\nabla_{a_i^t}\log\mu_t^i(a_t^i|s)\nabla_{\theta_i}a_i^t|_{a_i^t=\alpha_t\pi_{\theta_i}(s)+\sigma_t\epsilon_i} \\
&\quad -\nabla_{a_i^t}\log\pi_t^i(a_t^i|s)\nabla_{\theta_i}a_i^t|_{a_i^t=\alpha_t\pi_{\theta_i}(s)+\sigma_t\epsilon_i}\big] \quad (20) \\
&= \mathbb{E}_{\boldsymbol{s}\sim\mathcal{D}^\mu}\nabla_{a_i}Q_\phi(\boldsymbol{s},\boldsymbol{a}_i,\boldsymbol{a}_{-i})|_{\boldsymbol{a}_i\sim\pi_\theta^i(\boldsymbol{s}),\boldsymbol{a}_{-i}\sim\pi_\theta^{-i}(\boldsymbol{s})}\nabla_{\theta_i}\pi_i \\
&+ \frac{1}{\beta}\mathbb{E}_{t,s,\epsilon_i}\frac{\sigma_t}{\alpha_t}\omega(t)\left[-\frac{\epsilon_i(a_i|s,t)}{\sigma_t}\alpha_t\nabla_{\theta_i}\pi_{\theta_i}(s) + \frac{\epsilon}{\sigma_t}\alpha_t\nabla_{\theta_i}\pi_{\theta_i}(s)\right] \\
&= \Bigg[\underbrace{\mathbb{E}_{\boldsymbol{s}}\nabla_{a_i}Q_\phi(\boldsymbol{s},\boldsymbol{a}_i,\boldsymbol{a}_{-i})|_{\boldsymbol{a}_i\sim\pi_\theta^i(\boldsymbol{s}),\boldsymbol{a}_{-i}\sim\pi_\theta^{-i}(\boldsymbol{s})}}_{\text{Q gradient}} \\
&\quad -\frac{1}{\beta}\mathbb{E}_{s,s,\epsilon_i}\omega(t)\Big(\underbrace{\epsilon_i(a_i^t|s,t)}_{\text{score }\mu_i^t} - \underbrace{\epsilon}_{\text{score }\pi_i^t}\Big)|_{a_i^t=\alpha_t\pi_{\theta_i}(s)+\sigma_t\epsilon_i}\Bigg]\nabla_{\theta_i}\pi_i(s)
\end{aligned}
$$

$\square$

## B.2 PROOF OF PROPOSITION 2

First, we derive the optimization objectives with centralized learning framework. By decomposing the KL term in (9), we have

$$
\mathcal{L}_{CTDE}^i = \mathbb{E}_{s\sim\mathcal{D}^\mu,\boldsymbol{a}\sim\pi_\theta(\cdot|s)}Q^{tot}(s,\boldsymbol{a}) + \frac{1}{\beta}\mathbb{E}_{s\sim\mathcal{D}^\mu,\boldsymbol{a}\sim\pi_\theta(\cdot|s)}\log\mu(\boldsymbol{a}|s) + \frac{1}{\beta}\mathbb{E}_{s\sim\mathcal{D}^\mu}\mathcal{H}(\pi(\boldsymbol{a}|s)),
$$

where $\mathcal{H}(\pi(\boldsymbol{a}|s))$ is the entropy of the joint policy. Then we need to distill the decentralized executive policy for each agent. Consider that each agent policy $\pi_{\theta_i}$ is an isotropic Gaussian policy, we can

decompose the joint policy by $\pi = \pi_{\theta_i} \pi_{\theta_{-i}}$. The gradient of agent $i$ is as follows

$$\nabla_{\theta_i} \mathcal{L}_{CTDE}^i = \nabla_{\theta_i} \mathbb{E}_{s \sim \mathcal{D}^\mu, \boldsymbol{a}_{-i} \sim \pi_{\theta_{-i}}(\cdot|s)} \left[ Q^{tot}(s, \boldsymbol{a}) + \frac{1}{\beta} \log \mu(\boldsymbol{a}|s) \right] \tag{21}$$

$$= \mathbb{E}_{s \sim \mathcal{D}^\mu, \boldsymbol{a}_{-i} \sim \pi_{\theta_{-i}}(\cdot|s)} \left[ \nabla_{\theta_i} Q^{tot}(s, \boldsymbol{a}) + \frac{1}{\beta} \nabla_{\theta_i} \log \mu(\boldsymbol{a}|s) \right] \tag{22}$$

$$= \mathbb{E}_{s \sim \mathcal{D}^\mu, \boldsymbol{a}_{-i} \sim \pi_{\theta_{-i}}(\cdot|s)} \left[ \nabla_{\theta_i} \pi_i * \nabla_{a_i} Q^{tot}(s, \boldsymbol{a}) + \frac{1}{\beta} \nabla_{\theta_i} \pi_i * \nabla_{a_i} \log \mu(\boldsymbol{a}|s) \right] \tag{23}$$

$$= \mathbb{E}_{s \sim \mathcal{D}^\mu, \boldsymbol{a}_{-i} \sim \pi_{\theta_{-i}}(\cdot|s)} \left[ \nabla_{a_i} Q^{tot}(s, \boldsymbol{a}) + \frac{1}{\beta} \nabla_{a_i} \log \mu(\boldsymbol{a}|s) \right] \nabla_{\theta_i} \pi_i. \tag{24}$$

Importantly, different from the cases in BRPO-Ind, we cannot distill a score function $\nabla_{a_i} \log \mu(\boldsymbol{a}|s)$ from the pretrained diffusion models of joint behavior policies. To illustrate the influence of inproporate factorizations, we slightly abuse the factorization assumptions to decompose the joint behavior policy as $\mu(\boldsymbol{a}|s) = \prod_{i=1}^n \mu_i(a_i|s)$ and propose a revised baseline called BRPO-FAC. This variant shares most of the framework with BRPO-CTDE, but differs in the policy regularization component: instead of using the joint behavior policy, BRPO-FAC employs individual behavior policies for regularization.

## B.3 PROOF OF PROPOSITION 3

We consider a fully-cooperative n-player game with a single state and action space $A = [0,1]^n$. Let $\pi^*$ be the optimal joint policy with two optimal modes: $a_1 = (1, \ldots, 1)$ and $a_2 = (0, \ldots, 0)$. Let $\hat{\pi}$ be a factorized approximation of $\pi^*$ such that $\hat{\pi}(a) = \prod_{i=1}^n \hat{\pi}_i(a_i)$, where each $\hat{\pi}_i$ is learned independently.

Given that $\pi^*$ has two optimal modes $(1, \ldots, 1)$ and $(0, \ldots, 0)$, and each $\hat{\pi}_i$ is learned independently, the best approximation for each individual policy is to assign equal probability to 0 and 1. Thus, each $\hat{\pi}_i$ converges to $\text{Uniform}(\{0, 1\})$, with $\hat{\pi}_i(0) = \hat{\pi}_i(1) = 0.5$ for all $i$.

Since each $\hat{\pi}_i$ is $\text{Uniform}(\{0, 1\})$, the joint policy $\hat{\pi}$ will have a mode for each possible combination of 0s and 1s across the $n$ players. There are $2^n$ such combinations. The probability of each mode is $\hat{\pi}(a) = \prod_{i=1}^n \hat{\pi}_i(a_i) = (0.5)^n = 2^{-n}$. Therefore, the reconstruction of joint policy $\hat{\pi}$ exhibits $2^n$ modes, each with probability $2^{-n}$.

To prove that the total variation distance between $\pi^*$ and $\hat{\pi}$ is $\delta_{TV}(\pi^*, \hat{\pi}) = 1 - 2^{1-n}$, we start with the definition of total variation distance:

$$\delta_{TV}(\pi^*, \hat{\pi}) = \frac{1}{2} \sum_a |\pi^*(a) - \hat{\pi}(a)|$$

For $\pi^*$, we have $\pi^*(a_1) = \pi^*((1, \ldots, 1)) = 0.5$, $\pi^*(a_2) = \pi^*((0, \ldots, 0)) = 0.5$, and $\pi^*(a) = 0$ for all other $a$. For $\hat{\pi}$, we have $\hat{\pi}(a) = 2^{-n}$ for all $2^n$ modes.

Calculating the sum of absolute differences:

$$|\pi^*(a_1) - \hat{\pi}(a_1)| + |\pi^*(a_2) - \hat{\pi}(a_2)| = |0.5 - 2^{-n}| + |0.5 - 2^{-n}| = 1 - 2^{1-n}$$

For the remaining $2^n - 2$ modes of $\hat{\pi}$:

$$\sum |0 - 2^{-n}| = (2^n - 2) \cdot 2^{-n} = 1 - 2^{1-n}$$

Therefore,

$$\delta_{TV}(\pi^*, \hat{\pi}) = \frac{1}{2} \cdot (1 - 2^{1-n} + 1 - 2^{1-n}) = 1 - 2^{1-n}$$

As $n \to \infty$, we have:

$$\lim_{n \to \infty} \delta_{TV}(\pi^*, \hat{\pi}) = \lim_{n \to \infty} (1 - 2^{1-n}) = 1 - \lim_{n \to \infty} 2^{1-n} = 1 - 0 = 1$$

This limit indicates a severe distribution shift between the true optimal policy $\pi^*$ and its factorized approximation $\hat{\pi}$ as the number of players increases.

### B.4 MOTIVATION OF SEQUENTIAL SCORE DECOMPOSITION

Consider the independent factorized $\pi(a_1, a_2|s) = \pi_1(a_1|s) \cdot \pi_2(a_2|s)$ and sequential decomposed joint behavior policies $\mu(a_1, a_2|s) = \mu_1(a_1|s) \cdot \mu_2(a_2|a_1, s)$, which are all parameterized.

From the definition of KL divergence, we have

$$D_{KL}(\pi(a_1, a_2|s)||\mu(a_1, a_2|s)) = \sum_{a_1, a_2} \pi(a_1, a_2|s) \log \frac{\pi(a_1, a_2|s)}{\mu(a_1, a_2|s)}.$$

Then the gradients with respect to parameters $\theta_1$ (parameterizing $\pi_1(a_1|s)$) and $\theta_2$ (parameterizing $\pi_2(a_2|s)$) are respectively:

$$\frac{\partial D_{KL}}{\partial \theta_1} = \sum_{a1, a2} \pi_1(a1|s) \cdot \pi_2(a2|s) \left( \frac{\partial}{\partial \theta_1} \log \pi_1(a1|s) - \frac{\partial}{\partial \theta_1} \log \mu_1(a1|s) \right),$$

and

$$\frac{\partial D_{KL}}{\partial \theta_2} = \sum_{a_1, a_2} \pi_1(a_1|s) \cdot \pi_2(a_2|s) \left( \frac{\partial}{\partial \theta_2} \log \pi_2(a_2|s) - \frac{\partial}{\partial \theta_2} \log \mu_2(a_2|s, a1) \right).$$

Clearly, such decomposition does not equate to that from KL divergences with independently factorized joint behavior policies, namely

$$D_{KL}(\pi_1(a_1|s)||\mu_1(a_1|s)), \ D_{KL}(\pi_2(a_2|s)||\mu_2(a_2|s, a_1)).$$

The difference is due to the "weighting" factors of $\pi_2(a_2|s)$ for $\frac{\partial D_{KL}}{\partial \theta_1}$ and $\pi_1(a_1|s)$ for $\frac{\partial D_{KL}}{\partial \theta_2}$, which account for the joint contribution of $a_1$ and $a_2$ in the original joint distribution.

## C COMPUTATIONAL RESOURSES

For `MAMujoco` experiments, we utilized a single NVIDIA Geforce RTX 3090 graphics processing unit (GPU). The experiments for running OMSD, OMAR, MA-DiffQL took 22H, 10H, 12H, for 2 agent environments, respectively. Note that the training time of OMSD contains two stages, 10 hours for pretraining diffusion models and 12 hours for training the MARL policies. For bandit experiments, it takes 10 minutes for each algorithm.

