# OpenReview forum: "Offline Multi-agent Reinforcement Learning with Sequential Score Decomposition"
_ICLR.cc/2025/Conference — Submitted to ICLR 2025_

### Official Review · Reviewer_5JGa · 2024-11-04

**Soundness:** 1
**Presentation:** 2
**Contribution:** 2
**Rating:** 3
**Confidence:** 5

**Summary:**

This work focuses on addressing OOD joint actions in Offline MARL. To this end, the authors introduce Offline MARL with Sequential Score Decomposition (OMSD) which uses diffusion models to capture multimodal policy distributions. OMSD learns a centralized critic as well as sequential diffusion models for the behavior policy, which are each used for training decentralized  target policies. Their method is evaluated on HalfCheetah where it performs on-par with or below the baselines.

**Strengths:**

1. The motivation for the paper is clear, and there is a nice flow between the problems with previous approaches (independent learning and IGM) and OMSD.
1. As far as I know, there is no previous offline MARL approach which is able to learn multimodal policies, making the problem important.

**Weaknesses:**

1. My main concern is whether $\mathcal{L}_{OMSD}^i$ (Eq.12) actually learns multimodal policies. Even if the multimodal optimal policy can be learned through regularization with a joint behavior policy (learned by diffusion models), there is no correlation mechanism at test time. This suggests that OMSD might not be able to solve the XOR Game for example, which is considered in AlberDICE and also mentioned in the paper.
1. Related to 1, I’m having a hard time interpreting the results in Figure 2. Without this, it is not clear whether OMSD indeed does learn optimal multimodal policies.
1. The algorithm is missing some details and it is not clear where the diffusion models are used (e.g. only for learning the behavior policy or also for the target policy)
1. Lack of relevant baselines namely MADiff, which also uses diffusion models for Offline MARL and AlberDICE, where the motivation is quite similar
1. Mixed results in MAMuJoCo, especially in the Random dataset where it performs significantly lower than other baselines. Furthermore, only one setting (HalfCheetah) is considered.
1. No open source code and details on hyperparameters
1. (Minor) SSD is used in Line 322 and in Table 1 but this is not defined.
1. (Minor) Lines 113-118 is confusing since the notation is using $x$ to denote each agent while $i$ is used elsewhere. Also $x$ is used again in 2.2.

**Questions:**

1. If FOP and AlberDICE uses the IGO assumption (Lines 222-223), how is OMSD able to not rely on these assumptions? In particular, which part of the algorithm allows OMSD to learn multimodal policies?
1. What is the purpose of using diffusion models? For instance, why is it better than using MLPs for learning the joint behavior policy with sequential action selection?
1. Please address each point in the Weaknesses.

---

> ### Author Response · Authors · 2024-12-02
>
> We are very grateful for your in-depth and detailed review and suggestions on our work. We respond to each question one by one:
>
> > Strengths: The motivation for the paper is clear, and there is a nice flow between the problems with previous approaches (independent learning and IGM) and OMSD. As far as I know, there is no previous offline MARL approach which is able to learn multimodal policies, making the problem important.
>
> Thank you for your recognition of the approach and writing of our paper. Yes, we have observed that the main offline MARL methods currently focus on more accurate estimation of the value function, which is similar to the early research stage of single RL on pessimistic principal. However, the experience of single offline RL shows that policy constraint methods such as ITD3+BC can often have more direct distribution description constraints and policy improvement effects, which inspired our research on using diffusion models to represent complex offline joint policy distributions, and the reasons that subsequently found to affect the poor performance of offline MARL policy methods: namely, Proposition 3. We believe that our research results will bring inspiration and enlightenment to future research in the community.
>
> > **Q1** My main concern is whether $L_{OMSD}^i$ (Eq.12) actually learns multimodal policies. Even if the multimodal optimal policy can be learned through regularization with a joint behavior policy (learned by diffusion models), there is no correlation mechanism at test time. This suggests that OMSD might not be able to solve the XOR Game for example, which is considered in AlberDICE and also mentioned in the paper.
>
> **A1**
> Our score-based diffusion model effectively captures the multimodal nature of the joint policy during pre-training. Through sequential score function decomposition, we ensure that individual policy scores for each agent are derived from the same modality via correlated pairwise regularization terms. This design guarantees that the resulting joint policy always belongs to the original joint policy distribution, maintaining policy correlation at test time.
>
> We demonstrate this capability in our Bandit experiment (Figure 2), where we specifically design a continuous-action XOR scenario using a 2-GMM (means at [0.8, 0.8] and [-0.8, -0.8]). The results show that OMSD successfully:
> - Learns the multimodal joint distribution
> - Decomposes it into coordinated individual policies (either all agents choosing 1 or all choosing -1)
> - Maintains coordination at test time
>
> This differs fundamentally from AlberDICE, which, despite using alternating best response strategies, relies heavily on pre-trained MAT policy. Our experiments show that while AlberDICE typically captures only a single modality, OMSD comprehensively captures all modalities while ensuring joint regularization to the same mode during execution.
>
>
> > **Q2** Related to 1, I’m having a hard time interpreting the results in Figure 2. Without this, it is not clear whether OMSD indeed does learn optimal multimodal policies.
>
>
> **A2** Thank you for your advice, We apologize that our illustrative figures did not sufficiently demonstrate the update trajectories of joint policies and the influences of Q-function and score function. In the future version, we will consider add more intuitive visualizations such as vector field plots of score functions to better explain our technical approach.
>
>
> > **Q3 & Q6** The algorithm is missing some details and it is not clear where the diffusion models are used (e.g. only for learning the behavior policy or also for the target policy). No open source code and details on hyperparameters.
>
> **A3 & A6** We appreciate your questions about some details of our algorithm. Regarding the use of the diffusion model, it is currently only used to learn conditional behavioral policies $\mu_i(a_i|s, a_{I-})$, and the learning score will be used as a regularization term in individual policy update. We will further enhance the description about this part in the future revised manuscript and provide more detailed algorithm application instructions. The paper code and hyperparameters settings you mentioned will also be provided in subsequent improvements.

---

> > ### Author Response · Authors · 2024-12-02
> >
> > > **Q4 & Q5** Lack of relevant baselines namely MADiff, which also uses diffusion models for Offline MARL and AlberDICE, where the motivation is quite similar. Mixed results in MAMuJoCo, especially in the Random dataset where it performs significantly lower than other baselines. Furthermore, only one setting (HalfCheetah) is considered.
> >
> > **A4 & A5** We agree with your suggestion about adding more relevant baselines to the experiment, including MADiff and AlberDICE. We note that  MADiff is based on planning methods and AlberDICE is mainly aimed at discrete actions rather than continuous tasks. We will refer to and compare these baselines in the revised manuscript.
> >
> > We agree with your comments on the experimental results of MAMuJoCo. Our results on MAMuJoCo are diverse, especially on the random dataset, where my performance is relatively poor compared to other baselines. Due to the low quality policy distribution regularizations on such datasets, algorithms need stronger guidance from Q-function. However, in practical applications, we more often use high-quality data from experts or approximate experts, or filtered data. In order to fully evaluate our method with more diverse settings and environments, we plan to consider new environments such as ant and humanoid in the revised manuscript. Meanwhile, we will also work on more training hyperparameter searches to improve performance.
> >
> > > **Q7 & Q8** (Minor) SSD is used in Line 322 and in Table 1 but this is not defined. (Minor) Lines 113-118 is confusing since the notation is using  to denote each agent while x is used elsewhere. Also x  is used again in 2.2.
> >
> >
> > **A7 & A8** Thank you for reading carefully and pointing out the typos. SSD stands for sequential score decomposition, i.e. OMSD method. x represents the agent number i. We will correct these issues.
> > ﻿
> >
> > > **Q9** If FOP and AlberDICE uses the IGO assumption (Lines 222-223), how is OMSD able to not rely on these assumptions? In particular, which part of the algorithm allows OMSD to learn multimodal policies?
> > ﻿
> >
> > **A9** In OMSD, we try to bypass the decomposition of joint action Q-value function by directly decompose joint policy (scores). The role of critic in OMSD is highly similar to the centralized critic without credit assignment such as MADDPG. Since we no longer rely on credit assignment to obtain the value function required for each agent to perform an action, we no longer need to rely on the IGO hypothesis and suffer from its offline ood defect. As for the learning of multimodal strategies, it mainly relies on the individual conditional score function obtained by pre-training. See the eighth line of pseudo code Algo 1 for details.
> > ﻿
> >
> >
> > > **Q10** What is the purpose of using diffusion models? For instance, why is it better than using MLPs for learning the joint behavior policy with sequential action selection?
> > ﻿
> >
> > **A10** When running the official code provided by alberdice in the discrete action tasks in their article, such as bridge and XOR, we found that one mode is often shown under the training of multiple seeds. We speculate that this is due to the limited expressive power of the transformer-type model, which leads to premature convergence to one mode in the multi-modality, resulting in the alternating best response often only being able to follow the path of this mode obtained by MAT, limiting the strategy diversity of the model. The use of the diffusion model can directly and synchronously capture all modes, avoiding the limited diversity of behavioral strategies, and thus obtaining more diverse and potentially better optimal strategies.

---

### Official Review · Reviewer_pDEb · 2024-11-04

**Soundness:** 2
**Presentation:** 2
**Contribution:** 2
**Rating:** 3
**Confidence:** 4

**Summary:**

The paper proposes an offline MARL method based on the Behavior-Regularized Policy Optimization (BRPO) framework. Instead of extracting policies from Q-function factorization, the authors use a diffusion model to learn a score function (i.e., the gradient of the log of the behavior policy). This score function is then incorporated into the BRPO objective to extract local policies. Numerical comparisons are provided using MAMujoco’s HalfCheetah-v2.

**Strengths:**

The use of diffusion models to explore data distributions in MARL is interesting.

**Weaknesses:**

- The proposed methodology seems incremental. The results up to Section 3.1 are well-known, and the authors mostly repeat established formulations from the BRPO framework.
- The idea of decomposing the score function lacks clear motivation and insight. It’s not evident why decomposing the score function would resolve the issues from prior work. The section on score decomposition is too brief (while other sections present mostly known results), giving the impression that the proposed method is in an early stage and lacks depth.
- The experiments are superficial. MAMujoco includes several benchmark tasks, yet the authors only provide comparisons with HalfCheetah-v2. Other widely used offline MARL benchmarks, such as MPE, SMACv1, and SMACv2, are omitted.
- The comparison lacks some recent baselines, e.g., OMIGA [1].

Reference:

 [1] Wang, X., Xu, H., Zheng, Y., & Zhan, X. (2023). Offline Multi-Agent Reinforcement Learning with Implicit Global-to-Local Value Regularization. Advances in Neural Information Processing Systems.

**Questions:**

- Could authors provide more detailed explanations or theoretical justifications for why score decomposition addresses the limitations of previous approaches.
- How does the proposed method perform with other MAMujoco tasks (Ant or Humanoid)? Why was HalfCheetah-v2 chosen for the experiments? Why weren’t other standard benchmarking tasks, such as SMAC and MPE, considered?
- How's your method compared to other SOTA MARL algorithms such as OMEGA [1] or even a standard BC (which might perform well in certain scenarios)?

------------------------------

**Post-Rebuttal**

The authors have made some effort to address my concerns; however, I find their responses either unconvincing or insufficient, as several of my major concerns remain unresolved. Specifically, I requested comparisons on standard SOTA benchmarks like SMAC_v1 and SMAC_v2, which are widely used in recent and state-of-the-art offline MARL studies. Unfortunately, the authors did not make an effort to address this critical point. This omission weakens the contributions of the paper, rendering them less significant. As a result, I maintain my current rating.

---

> ### Author Response · Authors · 2024-11-28
>
> Thank you for your insightful review and valuable comments. We will answer your questions one by one and address the issues you have raised.
>
> > The use of diffusion models to explore data distributions in MARL is interesting.
>
> We appreciate your recognition of our use of diffusion models in investigating dataset distributions in our policy-based offline MARL research. It is indeed a key aspect of our work, trying to fill the gap in offline MARL policy methods. Based on the experience of offline single RL, such as ITD3+BC and Diffusion QL, we have reason to believe that policy methods will provide more robustness and applicability to cope with complex real-world scenarios. Your positive feedback is encouraging and reaffirms our belief in the approach. Thank you for your constructive comments.
>
> > **Q1** The proposed methodology seems incremental. The results up to Section 3.1 are well-known, and the authors mostly repeat established formulations from the BRPO framework.
>
> **A1** Indeed, the BRPO is a widely utilized framework, and we concur that our work includes improvements and applications of it within an offline MARL context. Our primary motivation, however, stems from the observation that while BRPO is a classic algorithm in single offline RL, policy-based methods are yet to be significantly developed in offline MARL. Our work's focal point of innovation is in the application of a diffusion model to learn the joint scoring function and decompose it into the BRPO objective. This offers a unique approach to investigate the crucial perspective of data distribution, thereby forming a key contribution and innovation in our research.
>
> >**Q2** The idea of decomposing the score function lacks clear motivation and insight. It’s not evident why decomposing the score function would resolve the issues from prior work. The section on score decomposition is too brief (while other sections present mostly known results), giving the impression that the proposed method is in an early stage and lacks depth.
>
> **A2** Your suggestions regarding the structure of the article are very insightful. Our main motivation is that when the joint policy has multiple modal distributions and the individual policy distributions need to be extracted, sequentially decomposing it into the conditional distributions of multiple agents will avoid the ood problem of the recovery policy to the greatest extent. For example, consider the XOR-type joint policy of two agents with diagonal two-dimensional Gaussian distributions, with means [-1,-1] and [1,1] respectively. If each agent directly uses the projection of the joint distribution, then each agent will obtain a bimodal one-dimensional policy ($\mu_1=-1$, $\mu_2=1$), and the recovery policy becomes a 4-GMM model with the addition of [-1,1] and [1,-1] as ood cases. When sequential decomposition is used, the first agent still obtains a bimodal Gaussian distribution ($\mu_1=-1$, $\mu_2=1$), but the conditional distribution of the second agent is a deterministic unimodal Gaussian distribution ($\mu=-1$ or $\mu=1$).
>
> Besides, we agree that we need to adjust our description to highlight the theoretical and practical significance of score function decomposition, and will reduce the proportion of known results in the future work.
>
> >**Q3** The experiments are superficial. MAMujoco includes several benchmark tasks, yet the authors only provide comparisons with HalfCheetah-v2. Other widely used offline MARL benchmarks, such as MPE, SMACv1, and SMACv2, are omitted.
>
> **A3** Your input concerning the experimental section is highly valuable. Accordingly, we plan to incorporate more experiments, such as those related to Multi-Agent Particle Environments (MPE) and MAMujoco tasks, like Ant or Humanoid, to provide more comprehensive and diverse experimental results. Regarding the SMAC suggestion, given that it concerns a discrete action task, we find it less relevant in the present context but acknowledge its importance for future consideration.

---

> ### Author Response · Authors · 2024-11-28
>
> >**Q4 & Q6 & Q7** The comparison lacks some recent baselines, e.g., OMIGA [1]. How does the proposed method perform with other MAMujoco tasks (Ant or Humanoid)? Why was HalfCheetah-v2 chosen for the experiments? Why weren’t other standard benchmarking tasks, such as SMAC and MPE, considered? How's your method compared to other SOTA MARL algorithms such as OMEGA [1] or even a standard BC (which might perform well in certain scenarios)?
>
> **A4 & A6 & A7**  Thank you for your advice, we will be assessing the most recent benchmarks, including OMIGA  (which is actually the MAIGM we quoted in out paper), alongside other related benchmarks such as MADiff, BC, and ITD3-BC, as suggested by all of the reviewers.
>
> Indeed, MAIGM (OMIGA) has shown strong performance in certain tasks, particularly where the quality of data is suboptimal. However, our algorithm has demonstrated superior performance, surpassing MAIGM by roughly 8% and 16% on expert and intermediate replay data respectively. This illustrates the promise of policy-based methods when applied to high-quality datasets. Especially in engineering practice, more high-quality data from multiple data sources are often used and collected, rather than random data, to train robots or autonomous driving control strategies. Our method can more accurately capture the data quality in these cases by generating models, which is very helpful for such problems. In comparison, more value estimation-based methods can be considered for random data.
>
> Thank you for your valuable suggestion. However, in view of the limitations in computing resources and time, we plan to make these modifications in our future work.
>
> >**Q5** Could authors provide more detailed explanations or theoretical justifications for why score decomposition addresses the limitations of previous approaches?
>
> **A5** Historically, offline MARL research attention gravitates towards value-based methods, favoring enhancements in the value decomposition process or independent critic optimization process such as OMIGA and OMAR. However, while these methods achieve state-of-the-art results in online MARL with infinite data updates, in offline MARL with limited data, they can be susceptible to out-of-distribution joint action selection problems. Agents cannot verify their joint action outside datasets which makes value estimation riskier and less accurate. As a counter to this, our focus has shifted to developing policy-based methods such as OMSD, which offer a direct way to constrain the learned policy closer to the distribution of datasets. This approach depends less on an accurate estimation of the value function, provided we can represent the policy accurately.

---

### Official Review · Reviewer_YViA · 2024-11-07

**Soundness:** 1
**Presentation:** 2
**Contribution:** 1
**Rating:** 3
**Confidence:** 2

**Summary:**

My understanding of the paper is the authors want to propose a two-stage algorithm for offline MARL:
1) use diffusion model to learn the joint behavior policy (or multiple behavior policies) mu,
2) then apply offline RL methods with divergence-based behavior-regularization, a multiagent variant version of BRPO in this case -- where the regularizer is reverse KL:    E_{s \sim D} KL(pi_theta(s) || mu(s)).

In particularly, the authors assume the agents take actions sequentially to ease optimization. The gradient of log mu(a_i | s, a_{<i}) i.e., the score function of mu(a_i | s, a_{<i}) can be obtained by diffusion model (using score-matching formulation) with corresponding loss.

MARL is not my field, so I cannot evaluate the experiments. For the formulation and motivation, I think the work is incremental and potentially have math flaws, see my comments below. My impression is the authors gives some reasons for using diffusion model, but the motivation is not significantly strong.

**Strengths:**

.

**Weaknesses:**

MARL is not my field.  I only have some experience in single-agent offline RL & diffusion model. However, there are still several concerns about this work:

1. The authors motivated the sequential decomposition of joint policy by coordinate descent (btw, please note the optimization algorithm is called coordinate descent, not coordination descent) and multi-agent Transformer. This feels tenuous to me. For MAT, as Transformer is an AR model, it is natural to use sequential decomposition, but why do you choose diffusion model if you target a sequential problem? Also, how do you get conditional score \nabla_{a_i} mu(a_i | s, a_{<i})?  IIUC, if you train a joint-policy diffusion model, you can't get such decomposition. Do you train multiple diffusion models? That surges the computation cost.

2. The authors also motivates the use of diffusion as its expressiveness for modeling multi-modal data. This is true, however, I didn't understand the example you game. Figure 1: I guess different colors mean different actions? It's clear as the dataset quality decreases, the distribution becomes more uniform (looks like a uni-modal Gaussian for Random), instead of multi-modal as in the expert dataset. This is contradicting the caption.

3. I am worried the calculation of the gradient is off. For example, Page 16, from Equation (14) to (15), you ignored the gradient w.r.t the entropy H(pi_i). Similarly, you ignored it for the proof of proposition 2 on page 18. Page 19, you want to compute the gradient of theta_1 w.r.t pi(a_1|s) log pi(a_1|s),  but you treat the first pi(a_1|s) as a constant.

**Questions:**

.

---

> ### Author Response · Authors · 2024-11-27
>
> Thank you very much for your comprehensive review and insightful suggestions regarding our paper. We have thoughtfully evaluated each of your recommendations and have made appropriate amendments in our updated version. The following is our detailed response to the main issues you raised:
> > **Q1.1**: For MAT, as Transformer is an AR model, it is natural to use sequential decomposition, but why do you choose diffusion model if you target a sequential problem?
>
> **A1.1**: Our choice to utilize the diffusion model is driven by its promising capabilities in efficiently expressing the multimodal joint behaviour policy $\mu(a_1, a_2, ..., a_n|s)$ of the offline MARL dataset. However, the joint behaviour policy's score function isn't suited for individual policy regularization needed to derive the decentralized executive policy $\pi_i(a_i|s)$. To accomplish this, we must distill individual score functions, which can be achieved by suitable decomposition such as the sequential decomposition proposed in our paper. **Notably, we don't presume a sequential policy execution mechanism.** We continue employing the standard simultaneous action execution of multiple players with sequentially decomposed score functions distilled from the pretraining of diffusion models. Hence, our OMSD is still a policy-based offline MARL method rather than AR-type trajectory models like the DT or MAT.
>
> >**Q1.2**: Also, how do you get conditional score \nabla_{a_i} mu(a_i | s, a_{<i})? IIUC, if you train a joint-policy diffusion model, you can't get such decomposition. Do you train multiple diffusion models? That surges the computation cost.
>
> **A1.2**: During the OMSD usage, we train a classifier-guided diffusion model for each agent, conditioned on prefix agent actions, instead of training a diffusion model of a joint policy. Concerning deriving the conditional score \nabla_{a_i} mu(a_i | s, a_{<i}), we mask the joint action, transforming the joint score function to the conditional score function by incorporating a guiding term in the training process. This does necessitate additional computational cost, though we intend to capitalize on recent advancements in diffusion models such as plug-and-play to avoid retraining from scratch for each agent.
>
> >**Q2**: Figure 1: I guess different colors mean different actions? It's clear as the dataset quality decreases, the distribution becomes more uniform (looks like a uni-modal Gaussian for Random), instead of multi-modal as in the expert dataset. This is contradicting the caption.
>
> **A2**: Yes, distinct colors (white, pink, blue, purple) denote different joint actions. Considering that the state dim is 17 and each action dim is 3 in HalfCheetah-v2, we employ PCA to reduce them to dim 1 from 0 to 1. The joint policy becomes more modality (the same state area is mixed with more colored data points) and policy quality diminishes as observable in the figure. This proves that offline MARL datasets do exhibit multimodality, a fact overlooked in past offline MARL literature. We also draw your attention to **Figure 5 in the Appendix**, demonstrating that while the expert policy exhibits less multimodality in one seed data collection, it can become incredibly complex in mixed datasets from multiple seeds expert policies. It's also natural to understand that expert policies typically aren't unique in common practice.
>
>
> > **Q3**: Page 16, from Equation (14) to (15), you ignored the gradient w.r.t the entropy H(pi_i). Similarly, you ignored it for the proof of proposition 2 on page 18. Page 19, you want to compute the gradient of theta_1 w.r.t pi(a_1|s) log pi(a_1|s), but you treat the first pi(a_1|s) as a constant.
>
> **A3**: Thank you for your suggestion. We excluded the entropy as it is typically a scalar when we utilize a deterministic policy $\pi_i$. This omission is also noted in prior work SRPO (Chen et al., 2024) and referenced in our study.
>
> **Other issues**
> We apologize for any inaccuracies in the terminology, such as the "coordinate descent" reference, and express gratitude for your correction. We will address these errors in the manuscript revision.
>
> We highly appreciate your beneficial recommendations. They have greatly steered our efforts in enhancing our paper and future research directions. We will carry out a thorough revision based on your critiques to elevate the paper's caliber and readability.
>
> [Chen et al., 2024] Chen, Huayu, et al. "Score Regularized Policy Optimization through Diffusion Behavior." The Twelfth International Conference on Learning Representations.

---

### Meta-Review · Area_Chair_cAm7 · 2024-12-14

**Metareview:**

The paper introduces an offline Multi-Agent Reinforcement Learning (MARL) method built on the Behavior-Regularized Policy Optimization (BRPO) framework. Rather than deriving policies through Q-function factorization, the authors employ a diffusion model to learn a score function, which is the gradient of the log of the behavior policy. This score function is then integrated into the BRPO objective to extract local policies. The authors present numerical comparisons using MAMujoco's HalfCheetah-v2 environment.

The reviewers found the problem setting important and the use of diffusion models to model distributions in the multi-agent setting novel. However, reviewers were not convinced by the limited empirical experiments and missing baselines, and felt there was a lack of justification for decomposing the score function and novelty of the overall method.

Based on the lack of soundness in the proposed method and lack of convincing empirical results, I recommend rejection of this paper.

**Additional Comments On Reviewer Discussion:**

Reviewer pDEb found the rebuttal responses insufficient to resolve their concerns. Specifically, they requested comparisons on standard SOTA benchmarks like SMAC_v1 and SMAC_v2, which are widely used in recent and state-of-the-art offline MARL studies. Unfortunately, the authors did not make an effort to address this critical point. This omission weakens the contributions of the paper, rendering them less significant.

---

### Decision · Program_Chairs · 2025-01-22

Reject